# The Medial Prefrontal Cortex and Fear Memory: Dynamics, Connectivity, and Engrams

**DOI:** 10.3390/ijms222212113

**Published:** 2021-11-09

**Authors:** Lucie Dixsaut, Johannes Gräff

**Affiliations:** Laboratory of Neuroepigenetics, Brain Mind Institute, School of Life Sciences, Ecole Polytechnique Fédérale de Lausanne (EPFL), 1015 Lausanne, Switzerland; lucie.dixsaut@epfl.ch

**Keywords:** mPFC, fear memory, connectivity, engrams

## Abstract

It is becoming increasingly apparent that long-term memory formation relies on a distributed network of brain areas. While the hippocampus has been at the center of attention for decades, it is now clear that other regions, in particular the medial prefrontal cortex (mPFC), are taking an active part as well. Recent evidence suggests that the mPFC—traditionally implicated in the long-term storage of memories—is already critical for the early phases of memory formation such as encoding. In this review, we summarize these findings, relate them to the functional importance of the mPFC connectivity, and discuss the role of the mPFC during memory consolidation with respect to the different theories of memory storage. Owing to its high functional connectivity to other brain areas subserving memory formation and storage, the mPFC emerges as a central hub across the lifetime of a memory, although much still remains to be discovered.

## 1. Introduction

The understanding of memory processes in the brain has been at the heart of neuroscience research for more than a century, and a lot has been discovered since. We now know about brain regions, circuits, cells, synapses, electrophysiological, and molecular properties that are important for specific aspects of memory formation and storage. However, we are still quite far from understanding how precisely memories are created, and how and where they are stored for years so that we can still, for example, remember the house we grew up in.

The hippocampal formation (HPC) was the first region to be linked with episodic memory formation, with reported clinical cases of anterograde amnesia following HPC damage [1]. This role was further confirmed in rodent models by observing specific neuronal activation at precise timepoints in the life of a memory, as well as by manipulating subsets of HPC neurons [2]. From decades of research into the mnemonic role of the HPC, it is now firmly established that the HPC is necessary for the encoding of new associative memories, including fear memories, as well as for their early recall. This role was recently supported by the discovery of engram cells in the HPC. These neurons are first activated during the initial learning phase, undergo enduring molecular changes (cellular consolidation), and are reactivated during the recall of that same memory. In addition, the artificial reactivation of this neuronal ensemble triggers recall [3,4,5].

At the same time, we now also know that other brain areas are necessary during the process of memory formation and storage. For instance, it has for a long time been established that older memories are no longer stored in the HPC, but rather rely on cortical areas. Specifically, the medial prefrontal cortex (mPFC, see Box 1 for definition) was reported to be necessary for the recall of remote fear memories [6,7]. These findings gave rise to the classical model of memory formation, which posits that the initial formation and storage relies on the HPC, and that the mPFC is only recruited for longer storage [8].

Surprisingly, the mPFC was recently identified as crucial already during the early phases of learning using molecular or optogenetic manipulations [9,10,11,12]. Moreover, an mPFC engram population was identified already at learning, which could be reactivated to regulate memory at later times [13,14]. These findings urge us to reconsider the classical model of memory formation, and to further investigate the more complex role of the mPFC during fear memory consolidation.

Box 1Nomenclature of the rodent medial prefrontal cortex and its subregions.Although prefrontal areas have been widely studied
and implicated in various brain functions and disorders, there is a
surprising lack of commonly accepted nomenclature and delineation of its
subdivisions. Rodent stereotaxic atlases, on which experimentalists rely the
most, are regularly updated as no consensus is found [15]. In the absence of clear landmarks to define
the mPFC, a lot is left to individual appreciation which can lead to apparent
incoherencies between studies and overall misinterpretation. Until a unified
nomenclature is accepted in the field, it is necessary that authors report
precise stereotaxic coordinates and explicitly define the brain region(s)
they study.For simplicity here, we will use the following
nomenclature for the 3 major subdivisions of the mPFC:
-the **Anterior Cingulate Cortex** (ACC), sometimes referred to as Anterior Cingulate Area, dorsal and ventral (dACA and vACA-Allen Brain Atlas) or Cingulate cortex area 1 and 2 (Cg1 and Cg2-Paxinos and Franklin).-the **Prelimblic Cortex** (PL) or Prelimbic Area-the **Infralimbic Cortex** (IL) or Infralimbic Area
Although the human PFC evolved to be relatively
bigger and more complex than the rodent PFC, notably with more clearly
defined subregions, homologies in embryological development, layer
organization, cell-type distribution and connectivity patterns advocate for
potentially shared functions. For an anatomical definition and a comparison
between human and rodent PFC, see Carlén, 2017 [16].
For a very detailed description of the cytoarchitecture of the mouse PFC, see
Van de Werd et al., 2010 [17], and for a
comparison between mouse reference atlases, see Le Merre et al., 2021 [15].


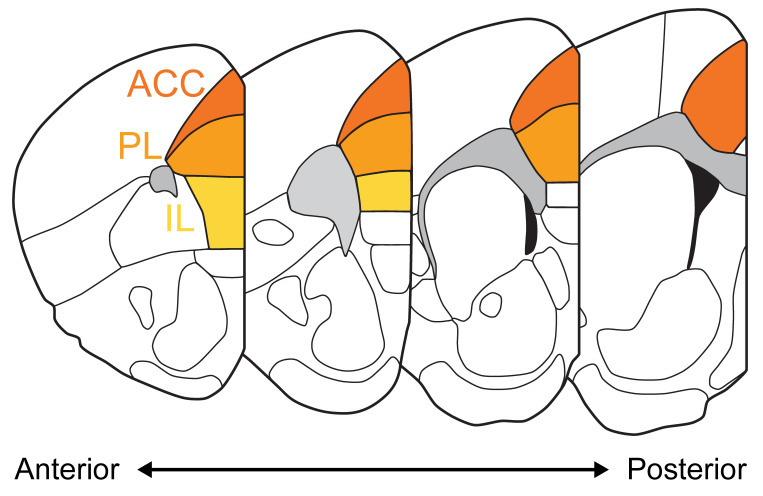

**Box figure.** Coronal sections of the mouse brain along the antero-posterior axis with the 3 major subdivions of the mPFC highlighted (Anterior cingulate cortex ACC, Prelimbic cortex PL, Infralimbic cortex IL) based on the Allen Brain Atlas.

In this review, we will summarize the available experimental data around the involvement of the mPFC in memory processes across consolidation, highlighting its importance during the entire lifetime of a memory. Then, we will consider the mPFC as a hub within a brain-wide network for memory storage and highlight its functional connectivity and engagement during this process. Lastly, we will discuss current theories of memory storage and how they can reconcile the experimental data within a global model of memory formation. Unless otherwise stated, we will focus on contextual fear memories, the most widely used rodent behavioral model to study memories that last up to several months. These salient memories are particularly interesting to investigate since they are often used in the context of traumatic memories in rodent and human studies alike.

## 2. The mPFC in Remote Memory Recall

The importance of the mPFC and its subregions has initially been described during the recall of old, or remote, memories. A memory is classically considered remote in rodents at least 12 days after the initial learning; if tested earlier, the memory is considered recent. In a seminal study using immediate early gene expression (IEG, see Box 2) as a marker of neuronal activity, the anterior cingulate cortex (ACC), the prelimbic cortex (PL), and the infralimbic cortex (IL) were shown to be activated specifically at remote and not recent recall following contextual fear conditioning (CFC) [7]. Moreover, inhibition of ACC, but not PL, with local lidocaine infusion prior to the recall prevented memory retrieval only at remote time points and not recent ones. Together with similar findings on spatial memories [6,18], these studies cemented an important role for the mPFC at remote times of a memory.

Box 2Neuronal activity visualization and engram-tagging techniques.Neuronal activity can be visualized using several
techniques with their unique advantages and disadvantages depending on the
research question being asked.**Electrophysiology** enables direct monitoring
of the electrical activity of a single neuron ex vivo or in vivo, with a high
temporal and spatial resolution, and can be used to identify specific
neuronal subtypes based on their distinct firing patterns [19,20]. At a lower spatial resolution, it is also
possible to record local field potentials or global electrical oscillation
patterns at the scale of an entire brain region [21,22].
**Calcium imaging** is based on a fluorescent
reporter, the activity of which correlates with intracellular calcium
concentration, considered as a proxy for neuronal activation [23]. It has a lower temporal resolution than
electrophysiology, but allows for monitoring of many neurons at once, with a
potentially high spatial resolution depending on the type of imaging
technique it is paired with it [24]. In both
cases, technological advances are increasing the power of those techniques to
monitor live brain activity in freely behaving animals with minimum tissue
damage [25,26].**Immediate early genes** (IEGs), such as cFos,
Arc, Npas4, Zif 268, etc., are transcribed upon neuronal activation.
Visualizing the corresponding mRNA or protein expressed after behavior
enables identification of recently active neurons. Taking advantage of the
specific dynamics of each of those IEGs, they can also be combined to
disentangle neurons that take part in two successive tasks [27,28,29,30], but they cannot provide permanent
labelling themselves.To this end, **conditional reporter expression
using IEG promoters** have then been developed, allowing for the
long-lasting tagging of neurons that were active at a given point in time.
Two main strategies have been used: (1) the **Tettag system** uses the
Doxycyclin-dependant transcription factor tTA, expressed under the cFos promoter,
to restrict the expression of a chosen protein under a Tet promoter to only
tag neurons active during the desired tagging time-window [31]; (2) the **TRAP system** uses a
Tamoxifen-dependent Cre recombinase Cre-ERT2 under a cFos or Arc promoter to
restrict recombination of a chosen gene to activated neurons at the time of
Tamoxifen injection [32,33]. In both cases,
those systems can be used within transgenic mouse lines or through
stereotaxic delivery of viral constructs. They allow restriction in time and,
if desired, in space, of the initial tagging, and subsequent manipulation of
engram neurons by either chemogenetic [34,35,36]
or optogenetic tools [37,38].More recently, **artificial promoters** have
also been designed to be more sensitive to neuronal activation, such as the
RAM promoters (Robust Activity Marking), while maintaining their sensitivity
to a given transcription factor cascade, for instance, cFos and Npas4 [39,40].Furthermore, **GRASP techniques** (GFP
Reconstitution Across Synaptic Partners) coupled with the Tettag system
enable visualization of direct synaptic contact between engram and/or
non-engram cells that are located in different brain areas, and specifically
manipulate those pair types [41].Lastly, using cell-type specific promoters or viral
vectors with restricted tropism or antero- and retrograde transport, the
scope of action of these techniques can be even more refined. These methods
can also be coupled to enhance their potential. For instance, calcium imaging
and Tettag tagging were used together to monitor replay of engram cells
during sleep [42].

Follow-up studies confirmed the importance of the mPFC for remote memories. Brain-wide cFos mapping following CFC recall not only found an increased activity of mPFC subregions at remote over recent recall, but also identified these cortical regions as central network hubs at remote times [43,44]. Correlating with the cFos increase in the ACC as the memory ages, spine density was found to increase specifically after remote recall, whereas in the CA1 region of the hippocampus (HPC), spine density only increased after recent recall [45,46]. In addition, precise electrophysiological recordings in ACC together with other brain regions demonstrated a brain-wide reorganization of the brain networks with the age of a memory, highlighting the predominant role of ACC at later time points [47,48]. Lastly, establishing the functional involvement of the mPFC, optogenetic inhibition of ACC during recall specifically impaired remote memory [49].

Taken together, these data suggest that the mPFC becomes predominantly important for the expression of old memories. Conjointly with findings showing that the HPC was primarily required at early recall [1,50,51], this led to and underscored the classical dogma of memory formation, called the standard theory of memory consolidation [8] (also see the discussion below).

## 3. The mPFC in Early Memory Phases

The restricted involvement of the mPFC to remote time points was, however, challenged by many studies over the years, and increasingly so as technical advances allow for a more precise dissection of neural circuits.

### 3.1. At Memory Encoding

The first hint suggesting that the mPFC is already involved early on originates from long-term potentiation (LTP) recordings after auditory fear conditioning (AFC) in rats: Doyère and colleagues demonstrated that fear memory encoding elicited LTP in the CA1 to PL projection already 20 min after training [52]. A decade later, a somewhat overlooked study observed that reversible ACC inhibition with a GABAA agonist at AFC encoding impaired recent memory recall, and that electrical stimulation paired with tone presentations resulted in the creation of an artificial fear memory to the tone [53]. These findings already suggested that the ACC is necessary to encode fear memories but was not followed up until later studies started examining the mPFC at encoding again.

Investigating IEG induction, brain-wide cFos quantification after encoding and recall of several conditioning protocols highlighted a significantly increased activity in many cortical areas at early time points, notably in all mPFC subdivisions at CFC encoding [54]. Furthermore, whole mPFC inhibition prior to encoding using either TetTox (tetanus-toxin light chain blocking synaptic transmission) or synaptotagmin-1 knockdown to reduce synaptic transmission resulted in altered context discrimination at recent recall, and impaired remote recall [55], suggesting an overall role of the mPFC at encoding that broadly impacts memory consolidation. In parallel, several studies focusing specifically on the ACC pointed out transcriptional changes at encoding [9] as well as necessary new spine development for proper memory consolidation [56]. Nevertheless, the requirement of protein synthesis in ACC after learning for recent recall is still debated [10,12]. However, most strikingly, optogenetic inhibition of the ACC during encoding prevented memory retrieval at both recent and remote time points [9].

In contrast, data on the involvement of the PL and IL at encoding are not as clear and yielded conflicting results. In one study, lesion of the IL using NMDA infusion long before CFC training impaired recent memory recall, but PL lesion did not [57]. This result was also observed in an active avoidance paradigm using electrolytic lesions in IL or PL [58]. On the contrary, inhibition of global protein synthesis 1 h after CFC encoding in PL impaired recent recall [12]. In another study, optogenetic activation of parvalbumin (PV) interneurons PL/IL before AFC training had no effect on recent recall, but, surprisingly, the activation of CaMKII neurons impaired recent memory recall to both context and tone [59]. Different techniques used to inhibit those regions, different stereotaxic coordinates, as well as slightly different behavioral paradigms and timing of the inhibitions could explain those disparities. Nevertheless, these studies highlight the importance of the mPFC and several of its subregions during the encoding phase of a fear memory.

### 3.2. At Recent Memory Recall

In addition to the encoding phase, the mPFC was also found to play a role during recent memory recall. For example, optogenetic activation of ACC to dCA1-3 projection led to increased freezing to a neutral context one day after CFC [11]. Inhibiting this projection at recent recall impaired memory, suggesting that the ACC has a top-down control over the HPC at recent recall. In a separate study, optogenetic inhibition of the whole PL during an AFC recall impaired freezing both at 6 h and 7 days post AFC, times at which cFos was already increased in PL [60]. Along the same lines, inactivation of PL with muscimol at recent AFC recall impaired freezing to the tone, whereas inactivation of IL specifically prevented extinction learning at recent recall [61]. Such early importance of the mPFC was also confirmed using electrophysiological recordings during an AFC recall, identifying the necessity of mPFC neuronal assemblies to be functionally organized in a precise oscillation phase in order to drive memory recall [62]. The mPFC is therefore also involved in early recall of conditioned fears, most likely with different roles of its subdivisions.

Together with evidence that the HPC is also involved for the recall of remote memories [49,63], the involvement of the mPFC at early time points provides supporting evidence for the multiple trace theory (MTT, see below).

## 4. The mPFC Engram

Further evidence regarding the engagement of the mPFC early in the lifetime of a memory surfaced with the development of precise engram technologies (see Box 2). Engram cells are commonly defined as “a population of neurons that are activated by learning, have enduring cellular changes as a consequence of learning, and whose reactivation by a part of the original stimuli delivered during learning results in memory recall” [5].

A first indication that a subset of mPFC neurons are carrying memory-related information at encoding came from a study by Zelikowsky et al. [64]. Although not considered an engram study *per se*, it is the first to investigate the reactivation of mPFC neurons between encoding and recent recall, using a catFISH technique with Arc mRNA (see Box 2). In both a classical CFC protocol and an immediate shock protocol, PL neurons were found to be strongly reactivated after recent recall. Moreover, a population of PL neurons encoded context information from encoding to recall, and an additional population specifically encoded the context–shock association. This significant reactivation of PL neurons linked to a learning event, but not IL neurons, suggested that PL could contain engram cells already at encoding, but the remaining criteria to define engram cells were still to be fulfilled.

In a seminal study using a Tettag engram-tagging approach (see Box 2), Kitamura and colleagues [13] showed that PL neurons activated and tagged at CFC encoding could be optogenetically reactivated at later time points and that this reactivation could drive freezing behavior in a neutral context at both a recent and a remote time point. However, using IEG as well as calcium imaging, they observed that PL neurons tagged at encoding were naturally reactivated only during a remote recall and not a recent one. In addition, the inhibition of the CFC-tagged neurons during a recent recall had no effect on freezing but impaired memory recall at a remote time point. This study suggested the idea of a “silent engram” in the mPFC, which is already activated early on but requires consolidation to be naturally recruited in order to drive recall.

These important findings were confirmed by two subsequent studies using different techniques. The first one used chemogenetics in a viral TRAP setting (see Box 2) to demonstrate that the inhibition of CFC-tagged PL neurons specifically impaired remote and not recent recall in a context-dependent manner, and that the activation of this population was able to drive freezing at both recent and remote time points in a neutral context [14]. In addition, CFC-activated PL neurons were only naturally reactivated at remote and not recent recall. The second study used an AFC paradigm and a transgenic TRAP system, and confirmed the time-dependent reactivation of PL neurons at recall and the ability of those cells to drive recall 1 day after conditioning [32].

Taken together, these results demonstrate that engram cells are present in PL from encoding but are kept silent during early recall until the full consolidation of the memory. The existence of engram cells in ACC and IL still needs to be investigated.

## 5. Post-Learning Molecular Modifications in the mPFC

Similar to findings in the hippocampus, *de novo* transcription as well as *de novo* protein synthesis are required in the mPFC at CFC encoding for proper memory consolidation [10,12,65]. More precisely, whole-ACC RNA-seq revealed a global shift in the mPFC transcriptome towards biological processes linked with synaptic plasticity, and a decreased expression of genes involved in immunity 1 h after conditioning [9]. At the scale of a neuron, CFC triggered an increase in spine density as well as in mini excitatory post synaptic potential (mEPSCs) frequency, and the number of docked vesicles in individual synapses increased. Overall, this suggested that 1 h after CFC, ACC neurons are globally primed for plasticity and activity. In addition, epigenetic modifications could also be important molecular changes resulting from learning as cortical areas, such as the orbitofrontal cortex, elicited histone H3 acetylation after learning for remote memory consolidation [66].

As parts of the whole region, we can hypothesize that engram cells would also share those molecular modifications after learning, but whether those are specific features of engram cells still remains unknown. Using a viral TRAP approach, PL engram cells were found to require functional CREB (cAMP response element binding) for proper consolidation. Indeed, expressing a repressor of CREB-dependent transcription in engram cells at encoding specifically impaired remote memory recall [14]. More precisely, single cell RNA-seq of mPFC engram cells tagged at remote recall identified specific transcriptional programs induced by consolidation, and dependent on neuronal subtypes [67]. Those specific transcriptional programs share some common features, such as upregulation of genes involved in vesicle exocytosis, suggesting an enhanced neurotransmitter release in engram cells.

Overall, these studies indicate early mPFC molecular and transcriptional changes, with potentially important roles in long-term memory consolidation.

## 6. The mPFC Functional Connectivity

The mPFC is a densely connected structure receiving and sending projections from all major areas in the brain. As such, it is not surprising that it was identified as a central hub by brain-wide cFos mapping during CFC learning and recall [43,44]. However, which subsets of cells within the mPFC were responsible for this hubness, as well as their direct functional connectivity to other brain areas during memory formation and recall was not investigated in these studies. In the following section, we will describe the anatomical projections to and from the mPFC that have been functionally implicated in memory formation and retrieval (see Figure 1 and Figure 2). For a detailed review on connectivity and subregions specificities, see Le Merre et al. [15].

### 6.1. mPFC Functional Inputs

Only few of the mPFC inputs were functionally investigated in the context of fear memory formation and retrieval, mainly from regions that were also implicated independently of those processes, namely the entorhinal cortex (EC), the BLA, and the HPC. More precisely, at the time of memory encoding, three inputs to the mPFC were demonstrated to be required for long-term learning (represented in blue in Figure 1): BLA to PL, MEC (medial part of the EC) to PL, and CA1 to ACC. In all cases, inhibition of those projections at the time of encoding (the BLA and MEC inputs to PL were inhibited using optogenetics [13], while the CA1 to ACC was inhibited using chemogenetics [68]) led to impaired remote memory recall, while recent recall remained unchanged. This suggests that CA1, BLA, and MEC all send crucial information to the mPFC at the time of encoding that is necessary to establish and consolidate a functional cortical engram.

At recent recall, two additional inputs were functionally characterized (represented in green in Figure 1): dCA1 to PL and IL, as well as BLA to mPFC. The dCA1 input was studied chemogenetically in an inhibitory avoidance (IA) paradigm. The results highlighted a differential effect on these two subregions, as the inhibition of the dCA1 to PL projection impaired memory expression, while inhibition of the dCA1 to IL pathway prevented extinction [69]. A similar dichotomy between fear recall and extinction was observed in the context of an AFC paradigm investigating the electrophysiological properties of BLA neurons projecting to mPFC but without distinction between mPFC subregions [70]. Two neuronal subpopulations encoding either fear or safety after extinction were identified, both projecting to the mPFC but part of distinct neuronal circuits: BLA fear neurons received input mainly from the ventral HPC and extinction neurons from the mPFC. Subsequent studies confirmed that fear and extinction neurons project to the PL or the IL, respectively [71,72,73].

Surprisingly, a functional input has yet to be identified at the time of remote recall, although the storage role of the mPFC after consolidation has long been established. However, an interesting experiment could give a hint on where to start looking. Kitamura et al. [13] investigated the inputs of mPFC engram cells tagged at CFC encoding using a Tettag system and monosynaptic retrograde tracing with a modified rabies virus. With this technique, PL engram cells were found to receive direct input mainly from anteromedial (AM) and medial dorsal (MD) thalamic nuclei, MEC (layer V), BLA, retrosplenial cortex (RSP), and vHPC, but also from the lateral orbital cortex, insular cortex, medial septum, subiculum, and posterior ACC. Future research on the functional inputs of mPFC engram cells should therefore focus on those specific long-range projections having a monosynaptic connection with engram cells.

### 6.2. mPFC Functional Outputs

The outputs of the mPFC involved in fear learning were mainly investigated in a correlative manner (i.e., a projection exists, and the target area is more or less activated at given times), and only few research articles have reported manipulations to assess the functional role of these projections. It is important to note that overall, engram cells as a population seem to project to the same brain regions as the mPFC as a whole. Indeed, in an experiment using TRAP transgenic mice, PL engram cells were tagged at the time of recent or remote recall and their axonal projections were traced in a brain-wide manner, highlighting that engram cells project to all the classical target regions of the mPFC [32].

At recent recall, three projections from the mPFC were studied at relatively early time points, to the HPC and the amygdala, as well as to the lateral habenula (LHb) (green in Figure 2). A direct mPFC to HPC projection was identified between the ACC and the dCA1-3, and was found necessary for a recent CFC recall by optogenetic manipulation [11]. In addition, the ACC to dCA1-3 projection recruited a subpopulation of HPC neurons that encodes contextual information, suggesting that the mPFC exerts top-down control over the HPC at the time of recent recall.

In parallel, the dense projections between the mPFC and the amygdala have also been studied at recent time points but using AFC protocols. Specifically, optogenetic inhibition of PL to BLA projection prevented memory expression 6 hours after AFC training but not 7 days later, suggesting that this projection is necessary for a recent recall [60]. The link between the mPFC and the BLA at this time point is also supported by oscillations studies, which, although independent of direct connectivity, confirm that the mPFC drives BLA neurons firing during a recent AFC recall via 4 Hz oscillations that synchronize the two regions [74]. Investigating the subregion connectivity in more depth, Knapska et al. [75] correlated lateral amygdala (LA) neuron activity in high or low fear states after AFC recent extinction with different inputs from the mPFC. In the non-extinguished context where rats showed high fear, activated LA neurons received input preferentially from the PL, and in the extinguished context where the animals were less afraid, they had more IL inputs. These findings were later expanded with chemogenetic inhibition of the IL to BLA projection, resulting in an impaired extinction [76]. This further confirms the dichotomy between PL and IL in fear and extinction, as well as the importance of mPFC input to the amygdala in recent AFC recall.

Finally, the LHb also receives specific input from the mPFC, and notably projections from a PL engram tagged in a recent recall of a shock-based conditioned place preference paradigm [77].

At more remote points, projections from the PL to the paraventricular thalamus (PVT, represented in red in Figure 2) were found to be necessary for a 7-day-old recall of a CFC, although not at a 6 h recall, suggesting a progressive recruitment of this cortico-thalamic connection during memory consolidation [60]. At a later remote recall, PL engram cells tagged at CFC encoding projecting to the BLA are also progressively recruited (represented in red in Figure 2). Indeed, optogenetic terminal inhibition during recall only impairs memory after 22 days but not before [13]. In parallel, electrophysiological recordings between ACC and CA1 highlighted the progressive synchronization of the two regions during the consolidation of a CFC memory, resulting in ACC coordinating CA1 neuronal activity via phase locking of the firing to the ACC theta rhythm at remote time points only [48]. Interestingly, the ability of ACC to drive CA1 activity was linked to successful memory recall, suggesting that at remote recall also, the mPFC could exert top-down control over the HPC.

Globally, the mPFC functional connectivity has only being partially investigated, and we still lack a comprehensive understanding of the neuronal network involved during fear memory consolidation. Surprisingly, the projections to and from the thalamic nuclei have been little studied, although some of these regions are becoming of high interest in the regulation of the consolidation and the extinction of both recent [78] and remote fear memories [79,80]. This, in addition to the possibility of tracing connections from and to engram cells specifically [13,32], opens up many areas of investigation that will be of high interest in the future.

## 7. Models of Memory Formation

Several models of memory formation, storage, and retrieval have been brought forward over the past decades that encompassed all the experimental results that were available at the time of writing. As techniques evolved and became more refined, these models were continuously updated and complexified. Here, we will summarize the main theories that emerged and try to reconcile them with the abovementioned experimental findings of the mPFC during the different phases of memory consolidation.

### 7.1. The Standard Model of Memory Formation and Its Limitations

The first model of memory formation, known as the standard or the classical model, relies on the early findings suggesting that the HPC was only necessary for the initial storage and early retrieval phase, while cortical areas and in particular the mPFC were later progressively recruited for the long-term storage and retrieval of old memories [6,7,44,46,50,51]. This model was proposed by Squire [81,82] based on extensive data gathered from humans, primates, and rodents, and was further refined by Frankland and Bontempi [8]. It was initially based on purely observational studies, then on broad inhibition studies using lesions or pharmacological inhibition, and more recently on IEG expression studies.

Several findings have in the meantime challenged this standard model. On the one side, cortical areas were shown to be involved in early phases of memory formation as well [9,10,53,54,55,56,71], and on the other side, the HPC was also found to be necessary for the recall of remote memories, and not only recent ones, by means of pharmacological [63] as well as optogenetic inhibition [49]. Precisely, an acute optogenetic inhibition of CA1 during remote recall impaired memory, but a prolonged inhibition that started before the recall itself did not impair memory and resulted in increased cFos expression in ACC, which suggested the recruitment of alternative cortical circuits when the HPC is inhibited for an extended period of time [49]. Conversely, opto- and chemogenetic reactivation of DG engram cells could alter freezing even at remote timepoints, indicating that the hippocampal engram stays functional even after consolidation [13,27]. Taken together, these results advocate for the existence of parallel memory traces distributed in different brains areas.

### 7.2. The Indexing Theory (IT)

The IT, initially developed by Teyler and DiScenna [83], theorized how the HPC would exchange information with cortical areas, notably the mPFC. The HPC is presented as an ideal storage place for maps, or indexes, of specific cortical locations, enabling a link to be made between individual cortical modules that were co-activated during the initial learning event. Upon recall, which would reactivate some of the cortical modules involved in the original memory trace, the HPC index would reactivate the full panel of cortical modules in order to recreate the complete original memory. No memory content would be stored in the HPC, only the location of cortical modules that as a whole would form the memory trace. Initially, this theory was developed as an explanation of the first phase of the standard theory model, when memories are still HPC dependent, and suggested that older consolidated memories would not necessarily rely on an HPC index anymore. More recently, the IT was revisited and expanded, including our current knowledge of engrams: DG engram reactivation experiments driving freezing point to engram cells serving as indexes themselves [3,4,13]. However, it still remains unclear if indexes are still present in the HPC after remote consolidation or if they disappear, as cortical connections between modules appear to be strengthened enough to not require HPC indexes anymore [84,85]. Interestingly, the presence of engram cells in the mPFC [13,14,32] could also be explained if their activation triggers the HPC indexes that in turn would drive memory recall.

### 7.3. The Multiple Trace Theory (MTT)

The MTT (sometimes referred to as the trace transformation theory) was proposed by Nadel and Moscovitch [86] in order to take into account the experimental data challenging the standard model, and was further developed more recently [87]. It incorporates the IT to a broader model, which also extends into remote memory consolidation. The MTT postulates that the hippocampal formation is the first to encode the initial information upon learning, in a sparse and distributed manner, and that, similarly as in the IT, this hippocampal ensemble acts as an index to link the cortical neurons that are also representing the incoming information. Both neuronal ensembles, cortical and hippocampal, form the full memory trace, or engram as we would call it nowadays. The hippocampus provides the contextual information to link the individual pieces of information stored in the cortex from the start. However, in contrast to the IT, a new hippocampal trace will be created at each recall, indexing the same, or part of the same, cortical information, resulting in the creation of parallel and multiple traces distributed in the brain, making them harder to disrupt or erase. This theory would explain how new schemas can be integrated faster into an existing framework that was previously learned [88]. It is supported by studies highlighting the role of the HPC at late times post-encoding [49,63], of the mPFC at early times [9,10,53,54,55,56,71], as well as by reports of distributed engram cells throughout the brain [31,39,89,90,91,92,93].

## 8. Memories Are Dynamic: Discussion and Outlook

Taken together, reconciling all experimental results in one unified theoretical model is difficult. A possible explanation of the apparent complexity of memory processes is their high flexibility and the fast dynamics of brain circuits. Indeed, many brain areas aside from the mPFC have been involved in learning and/or retrieval, which could potentially take over if one important brain circuit is damaged or impaired. For example, engram cells have been found in the HPC [3,39,89,90] and amygdala [31,91] but also in other cortical areas, such as the RSP. Interestingly, the RSP has been involved in the retrieval of remote fear memories [92], and early engram cells formed at the time of learning were found to drive fear recall [93] and correlate with spatial memory retention [94]. This suggests that cortical regions other than the mPFC can also store memories or at least parts of it, complicating the experimental deciphering of brain networks involved. Brain-wide (engram) studies and multiple-site recordings would enable taking into account the various regions implicated in these processes.

A consequence of this distribution of the memory network is that brain circuits can be redundant and therefore compensate for any natural or artificial perturbation of the system, allowing for the great stability of salient memories. Several experiments indeed reported such a reorganization of brain circuitries, notably of mPFC taking over from the HPC in case of lesion before CFC encoding [57] or prolonged optogenetic inhibition at the time of recall [49]. In addition, the dynamics of consolidation could be modulated bidirectionally by either adding a novel learning event between the encoding and the recall of a CFC [95,96] or during a spatial learning task [97], which resulted in a faster memory dependance on cortical areas, or conversely by adding external factors, such as odors, to prolong the HPC dependency of a memory [98]. Moreover, the phenomenon of co-allocation of memories that are experienced close in time underlines the importance of the brain state at the time of learning [91,99].

From a technical viewpoint, neuronal activity-based mapping by means of IEGs is also not without caveats, as the wide use of IEGs in engram-tagging strategies biases studies towards a specific neuronal population, e.g., those expressing cFos, while leaving aside other neurons in which activity would trigger a different transcriptional pathway [39]. Additionally, although neurons are thought to be the central elements in memory consolidation, it is now apparent that other cell types should also be taken into account as they can greatly influence memory formation, such as astrocytes in CA1 to ACC communication at encoding [68], HPC microglia in forgetting [100], and myelinating oligodendrocytes in the mPFC for remote memory consolidation [101]. Indeed, some of these cell types in the mPFC are showing important transcriptional modifications following remote recall [67], suggesting that they play a specific role in this process.

These limitations and open questions notwithstanding, it is clear from experimental data and the diverse theoretical models alike that the mPFC plays a critical role in all phases of the lifetime of a memory, the current state of the knowledge we have summarized in this review.

## Figures and Tables

**Figure 1 ijms-22-12113-f001:**
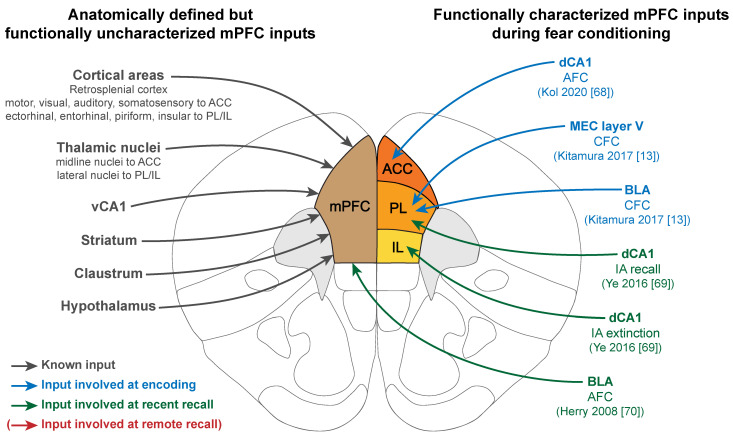
mPFC inputs are involved in encoding and recent recall of fear memories. Left: Anatomically defined inputs to the mPFC. Right: mPFC inputs functionally involved during fear memory consolidation, color coded by the phase of the memory (blue: encoding, green: recent recall, red: remote recall). Note the absence of functionally characterized mPFC inputs at the time of remote recall. AFC: Auditory fear conditioning; CFC: Contextual fear conditioning; IA: Inhibitory avoidance; MEC: Medial entorhinal cortex; BLA: Basolateral amygdala.

**Figure 2 ijms-22-12113-f002:**
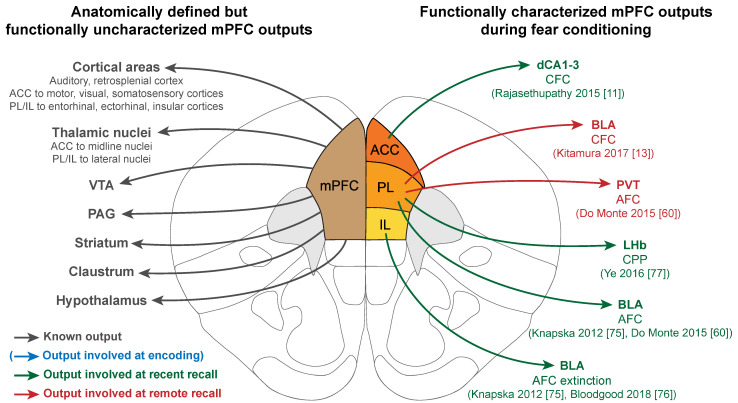
mPFC outputs are involved in recalling fear memories. Left: Anatomically defined outputs from the mPFC. Right: mPFC outputs functionally involved during fear memory consolidation, color coded by the phase of the memory (blue: encoding, green: recent recall, red: remote recall). Note the absence of functionally characterized mPFC outputs at the time of encoding. AFC: Auditory fear conditioning; CFC: Contextual fear conditioning; CPP: Conditioned place preference; BLA: Basolateral amygdala; LHb: Lateral habenula; PAG: Periaqueductal gray; PVT: Paraventricular thalamus; VTA: Ventral tegmental area.

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
