# Peer review of "The Medial Prefrontal Cortex and Fear Memory: Dynamics, Connectivity, and Engrams"

_ijms, 2021, doi:10.3390/ijms222212113_

Round 1
Reviewer 1 Report
The present manuscript offers a well-organized review of the neural bases of memory, highlighting the emerging role of the mPFC in the various aspects of memory encoding, formation, and recall. Along the way, a neat summary of memory formation models are provided, which would appeal to a broad readership. I appreciated the focus on the structural and functional connectivity of the mPFC with the rest of the brain in the context of memory – which offer useful insights for future research endeavors. Overall, I found the review to be timely and very well written. I just have a few minor suggestions.
One small improvement that could be made is to more clearly discuss the rationale for choosing fear memory for the review. The introduction begins with classic HPC lesion patient studies with episodic memory deficits, and towards the end jumps to contextual fear memories.
Another minor point, but I believe the readers would appreciate it if the newer models described in sections 5.2 and 5.3 could be explained by referring to empirical studies mentioned in the previous sections (a la 5.1).
Again, this is a very nice review of an important topic. Thank you for the opportunity to review this work.
Author Response
Thank you very much for your feedback. Here are our answers to your comments.
- One small improvement that could be made is to more clearly discuss the rationale for choosing fear memory for the review. The introduction begins with classic HPC lesion patient studies with episodic memory deficits, and towards the end jumps to contextual fear memories.
We updated the introduction so that the specific focus on fear memories in general and contextual fear memories in particular is clearer. These memories are particularly salient and therefore straightforward to study, but also give insights in the field of traumatic memories that are highly relevant in humans. For those reasons, they are widely studied and a lot of literature is available.
- Another minor point, but I believe the readers would appreciate it if the newer models described in sections 5.2 and 5.3 could be explained by referring to empirical studies mentioned in the previous sections (a la 5.1).
We added more references that support the IT and MTT theories, to give more weight to these theories as we did for the standard model.
Reviewer 2 Report
This is a well-executed review concerning the relationship between medial prefrontal cortex and fear memory. However, there are a few things to note:
1) anatomically it is more correct to use the term hippocampal formation (dentate gyrus, hippocampus proper and subiculum) than hippocampus;
2) Relationships involving prefrontal cortex and amygdala should also take into account the numerous scientific works of Joseph LeDoux's working group
Author Response
Thank you very much for your feedback. Here are our answers to your comments.
- Anatomically it is more correct to use the term hippocampal formation (dentate gyrus, hippocampus proper and subiculum) than hippocampus
We agree with the reviewer that the use of the term “hippocampal formation” is more correct than “hippocampus”. We have modified it in the manuscript.
- Relationships involving prefrontal cortex and amygdala should also take into account the numerous scientific works of Joseph LeDoux's working group
We agree with the reviewer that the work of Joseph LeDoux shed considerable light on the relationship between the cortex and the amygdala. In particular, his investigations are crucial in the field of aversive conditioning, where they mainly focus on the auditory cortex and auditory thalamus projections to the amygdala. As an example thereof, we have added reference [31] (Moscarello and LeDoux, J.Neurosci. 2013), which broadens the findings in contextual fear conditioning – the primary focus of this review – to an active avoidance paradigm.